# Association between Physical Exercise and Glycated Hemoglobin Levels in Korean Patients Diagnosed with Diabetes

**DOI:** 10.3390/ijerph19063280

**Published:** 2022-03-10

**Authors:** Il Yun, Hye Jin Joo, Yu Shin Park, Eun-Cheol Park

**Affiliations:** 1Department of Public Health, Graduate School, Yonsei University, Seoul 03722, Korea; ilyun94@yuhs.ac (I.Y.); hjjoo22@yuhs.ac (H.J.J.); dbtls0459@yuhs.ac (Y.S.P.); 2Institute of Health Services Research, Yonsei University, Seoul 03722, Korea; 3Department of Preventive Medicine, Yonsei University College of Medicine, Seoul 03722, Korea

**Keywords:** diabetes, glycated hemoglobin, HbA1c, physical exercise, walking exercise, resistance exercise

## Abstract

This study aimed to identify the association between physical exercise and glycated hemoglobin (HbA1c) levels in Korean patients diagnosed with diabetes. Data from the 2015–2019 Korea National Health and Nutrition Examination Survey were used. In total, 2559 participants were included (1286 males and 1273 females). Multiple logistic regression analysis was conducted to examine the effect of physical exercise on controlled HbA1c levels among diabetic patients. In Korean male patients with diabetes, performance of physical exercise, including walking and resistance exercises, was associated with controlled HbA1c levels < 6.5% (odds ratio (OR), 1.85; 95% confidence interval (CI), 1.17–2.92). In males, performing resistance exercise for ≥5 days a week, without walking exercise, had a significant association with HbA1c levels (OR, 1.75; 95% CI, 1.15–2.65). HbA1c levels were more likely to be controlled when both walking and resistance exercises were performed for ≥5 days a week in both sexes (males: OR, 1.74; 95% CI, 1.04–2.93 and females: OR, 2.59; 95% CI, 1.09–6.15). This study found that resistance exercise may contribute to the management of HbA1c levels among Korean patients with diabetes. Promoting resistance exercise performance can be beneficial for improving the condition of patients with diabetes.

## 1. Introduction

Since the prevalence of diabetes keeps increasing worldwide and related chronic complications cause higher health costs, primary prevention and management from the pre-diabetes stage are necessary [1,2]. Currently, diabetes is prevalent in one out of seven Korean adults aged > 30 years and it has been reported to be the sixth leading cause of death in South Korea [3]. Moreover, among Korean patients, 53.2% have been reported to be obese, with 61.3% and 72% of those with hypertension and hyperlipidemia, respectively. However, one-third of them were unaware of their diabetes status and <30% were being properly managed [4]. Early detection and management of diabetes and related complications have become major health policy issues that should be resolved at regional and national levels. As part of the policy, the Committee of Clinical Practice Guidelines of the Korean Diabetes Association has regularly published and updated the guidelines for Korean patients with diabetes [5].

Glycated hemoglobin (HbA1c) is a criterion for diagnosing diabetes along with fasting blood sugar, and good glycemic control is essential in preventing diabetic complications [6]. In general, diabetes is a progressive disease, which is not easy to improve or cure once diagnosed. Thus, control of blood glucose and HbA1c has to be achieved through lifestyle modifications, including diet and exercise, in addition to medication and insulin injection [7]. As a daily effort, it is recommended that patients with diabetes perform ≥ 150 min of moderate-intensity aerobic exercise per week and at least two sessions of resistance exercise per week [5,8].

Exercise training is an established part of the treatment in diabetes, and a number of preceding clinical trials have demonstrated that structured exercise training improves glycemic control and cardiovascular disease risk factors [9,10]. Several exercise trials have showed an overall reduction in HbA1c of 0.8% [11]. Some meta-analyses [12,13,14] also reported that structured exercise training has effects on glucose control and risk factors for complications. It has further been identified that combined aerobic and resistance exercise may improve the control of HbA1c more efficiently than aerobic or resistance exercise alone [15]. 

In this background, this study aimed to identify the association between physical exercise and HbA1c levels in Korean patients diagnosed with diabetes, and provides specific information on the type and duration of exercise required for controlling HbA1c levels. Furthermore, it aimed to provide health policy implications by examining whether diabetes can be improved if carefully managed through lifestyle modifications.

## 2. Materials and Methods

### 2.1. Data

The data used in this study were obtained from the 2015–2019 Korea National Health and Nutrition Examination Survey (KNHANES), a cross-sectional and nationwide survey conducted by the Korea Disease Control and Prevention Agency. The KNHANES was designed to evaluate the health status, health behavior, and nutritional status of the South Korean population to provide basic data for developing nationwide health policies [16,17].

Informed consent was obtained from all respondents in advance and all data analyzed in this study were fully anonymized. As the KNHANES complies with the Declaration of Helsinki and provides publicly accessible data, further ethical approval for the use of these data was not required [17,18].

### 2.2. Study Population

The total survey population from the recent five years (2015–2019) included 39,759 individuals. The inclusion criteria were as follows: those with self-reported diabetes diagnosed by doctors (questionnaire item: “Have you been diagnosed with diabetes?” and further questions about treatment history) and those aged >19 years. After excluding those with missing data (N = 367), data from 2559 participants (1286 males; 1273 females) were analyzed for this study. 

### 2.3. Variables

The dependent variable was the percent HbA1c, the optimal standard for monitoring glycemic control that could serve as an indicator of diabetes [19]. Diabetes was diagnosed when the HbA1c level was found to be ≥6.5% in a blood test. The main independent variable was physical exercise performance, which was subdivided into three groups: (1) people who performed only walking exercise; (2) people who performed only resistance exercise; and (3) people who performed both walking and resistance exercise. This variable was derived from questionnaires asking how many days each of walking and resistance exercise during the past week (0 to 7 days), and how long did your usually exercises at one time. According to the guidelines of the KNHANES, the rate of performing walking exercise was defined as the percentage of people who performed walking exercise for >30 min at a time, >5 days a week in the last week, and the rate of performing resistance exercise was defined as the percentage of people who practiced exercises, such as push-ups, sit-ups, dumbbells, barbells, and iron bars for >2 days in the last week.

The covariates included demographic factors (sex, age, marital status, and educational level), socioeconomic factors (income, occupation, and region), health-status related factors (body mass index [BMI] and the presence of other chronic diseases), and health behavioral patterns (drinking, smoking, and management of blood glucose). To control for the severity of diabetes, we adjusted the variables related to the management of blood glucose, which were subdivided into the following five categories: (1) those who received non-medication treatment; (2) those who received only oral medication; (3) those who received only insulin injection; (4) those who received both medication and insulin injection; and (5) those who did not manage their blood glucose at all. Those receiving both medication and insulin injection were considered as patients with the most severe diabetes.

### 2.4. Statistical Analysis

A descriptive analysis was performed to investigate the distribution of the general characteristics of the study population. Thereafter, multiple logistic regression analysis was conducted to examine the effect of physical exercise on HbA1c levels among patients with diabetes and to perform subgroup analyses stratified by sex, type of physical exercise, and HbA1c levels. The key results are presented as odds ratios (ORs) and 95% confidence intervals (CIs). For all analyses, we used SAS version 9.4 (SAS Institute Inc; Cary, NC, USA) and a *p*-value < 0.05 was considered statistically significant.

## 3. Results

Table 1 presents the results of univariate analyses, which examined the association between physical exercise and HbA1c levels with each variable stratified by sex. Of the total 2559 participants, 1286 were male and 1273 were female. Participants who self-reported that they had performed at least one day of walking and resistance exercises in the last week included 47.7% of males (N = 614) and 38.6% of females (N = 492). It was found that most participants had performed only walking exercise and the number of females that practiced resistance exercise was significantly lower than males.

Table 2 demonstrates the associations between physical exercise and other covariates and controlled HbA1c levels. Males who practiced both walking and resistance exercises in the past week were more likely to maintain HbA1c levels < 6.5% (OR, 1.85; 95% CI, 1.17–2.92). When walking and resistance exercises were combined, the ORs increased for both males and females, although not statistically significant.

We also performed subgroup analysis to assess the combined effects of physical exercise and other covariates on HbA1c levels, as shown in Table 3. It was found that when patients with obesity (BMI ≥ 25 kg/m^2^) performed both walking and resistance exercises, there was a significant lowering effect on the HbA1c levels (males: OR, 2.12; 95% CI, 1.06–4.25 and females: OR, 2.89; 95% CI, 1.15–7.27). In addition, males who drank frequently had a higher OR (OR, 2.33; 95% CI, 1.29–4.19).

Additional subgroup analysis was conducted to examine how the ORs changed according to the type and duration of exercise, that is, to investigate what kind of exercise and how long it needs to be performed to get a positive effect on diabetes. Duration of exercise performed in the last week was divided into four categories: never, 1–2 days a week, 3–4 days a week, and >5 days a week. As shown in Figure 1, for males, the ORs showed a tendency to increase linearly as the duration of resistance exercise increased. When the male participants performed exercise for >5 days a week, they were more likely to maintain HbA1c levels <6.5%, compared with those who did not perform resistance exercise (OR, 1.75; 95% CI, 1.15–2.65). Further, controlled HbA1c levels were associated with a combination of walking and resistance exercises for ≥5 days a week in both sexes (males: OR, 1.74; 95% CI, 1.04–2.93 and females: OR, 2.59; 95% CI, 1.09–6.15). This duration of exercise exceeds the global recommendations.

## 4. Discussion

Physical inactivity, obesity, and sedentary lifestyle are well-known important factors influencing the increasing incidence of diabetes worldwide [20,21]. It has also been found that regular exercise performance has a significant effect on nutrient metabolism. In particular, it helps to reduce blood glucose levels, improve glycemic control, and plays a role in weight loss [22,23]. In addition, weight and visceral fat loss, through exercise, may lead to improvement in metabolic indices and reduction in insulin resistance [24]. Consequently, for patients with pre-diabetes, performing exercise regularly may prevent the development of diabetes.

Based on this mechanism, our aim in the present study was to investigate the association between physical exercise and controlled HbA1c levels among Korean patients with diabetes. We found that a combination of physical and resistance exercises may contribute to the reduction of HbA1c levels and improve the condition of Korean male patients. Diabetes, a progressive and chronic disease, is difficult to improve or cure once diagnosed. However, our findings have suggested the possibility of improving glycemic control by performing physical exercise, especially resistance exercise. In other words, proper management through regular exercise, including resistance exercise, may be necessary from the early stage to prevent the condition from worsening.

Statistically significant results were observed only for males, but higher ORs were observed for both males and females when exercises, including resistance exercises, were practiced. Additionally, even if walking exercise was performed for >150 min a week (or >5 days a week for 30 min at a time), based on the global guidelines, without resistance exercise, no significant association was observed. On the other hand, if males performed only resistance exercise for ≥5 days a week, it was effective in managing their HbA1c levels, without walking exercise. Therefore, we suggest that performing resistance exercise is important and beneficial for improving the condition of patients with diabetes.

Additional subgroup analyses of the type and duration of exercise confirmed that there was a significant association in both sex groups when they performed exercise at higher levels than that officially recommended. Considering that the rate of exercise performance rarely increases due to changes in lifestyle, it is necessary to adjust the recommended exercise level for patients with diabetes, using both official national and regional guidelines. Additionally, considering that more than half of Korean patients with diabetes do not perform exercise, it is necessary to develop a program that allows easy access to exercise in daily life. 

Previous studies with a similar purpose to that of our study have shown that walking exercise duration of >150 min per week was associated with a reduction in HbA1c levels [7,14,25]. As in this study, the effect on controlling HbA1c was evaluated by dividing walking and resistance exercise, and it was demonstrated that a combination of walking and resistance exercise was more effective [15,22]. However, in the present study, statistical significance was found only in combined exercises and only among males; ORs were slightly increased in females as well. Therefore, we pointed out that the number of participants who performed exercise was too small. Despite the data being used in the last 5 years, among all female patients with diabetes, the rate of exercise performance, including resistance training, was only 8.09%. Therefore, given the observational nature of this study, the low rate of physical exercise performance among females, including the imprecise 95% CIs for some point estimates, the results of this study should be interpreted cautiously.

This study had certain limitations. First, because this was a cross-sectional survey, the association between variables could be confirmed, but causality could not be determined. Second, the data were based on self-reporting; hence, the actual level and duration of exercise reported may not have been accurately measured and may not be reliable. In order to provide a more reliable exercise therapy for HbA1c control in diabetic patients, future research should be conducted using wearable technology-equipped devices. Further, although we tried to adjust for numerous covariates that may affect the dependent variable, residual confounding effects from unmeasured variables could not be ruled out. For example, it is known that diet is as important as exercise in diabetes management [23], but there were no reliable diet-related variables from the extracted data, so it could not be corrected.

In Korea, only a few studies used recent data to determine this association. Findings derived from using past data might reflect a considerable level of discrepancy with the current lifestyle [26,27,28,29,30], and these studies did not provide patients with specific information on how to practice physical exercise. Therefore, our study is meaningful in that it reflected the current lifestyle patterns of Korean patients with diabetes, using the most recent and nationally representative data. Further, this study suggested that more intense exercise than that of the global recommendations should be implemented in intervention.

## 5. Conclusions

This study found that regular physical exercise was associated with controlled HbA1c levels among Korean patients with diabetes. Particularly, practicing resistance exercise may contribute more to the management of their HbA1c levels than walking exercise. Promoting resistance exercise performance can be beneficial for improving the condition of patients with diabetes.

## Figures and Tables

**Figure 1 ijerph-19-03280-f001:**
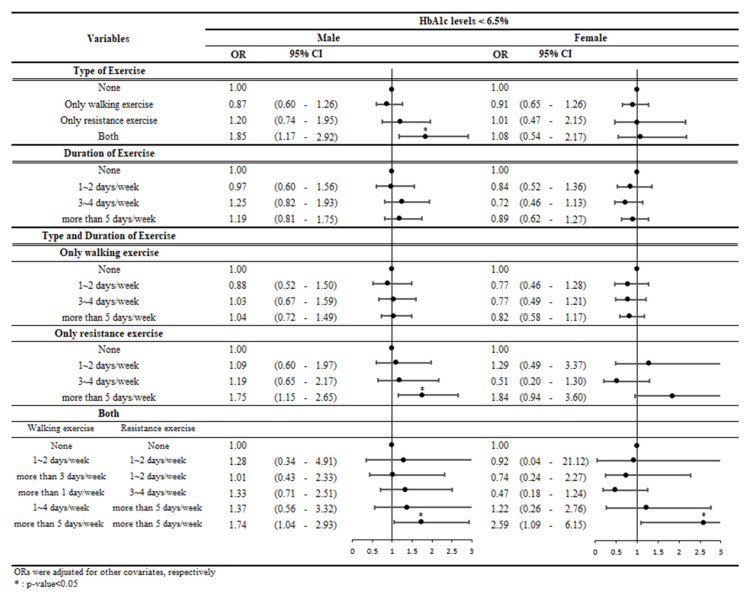
Results of subgroup analysis stratified by the type and duration of physical exercise. Abbreviations: HbA1c, glycated hemoglobin; OR, odds ratio; CI, confidence interval.

**Table 1 ijerph-19-03280-t001:** General characteristics of the study population.

Variables		Male							Female						
		HbA1c Level							HbA1c Level						
		TOTAL		≥6.5%		<6.5%		*p*-Value	TOTAL		≥6.5%		<6.5%		*p*-Value
		N	%	N	%	N	%		N	%	N	%	N	%	
Total (N = 2559)		1286	100.0	910	70.8	376	29.2		1273	100.0	912	71.6	361	28.4	
Physical exercise								0.109							0.619
	Yes	614	47.7	426	69.4	188	30.6		492	38.6	357	72.6	135	27.4	
	Only walking exercise	302	23.5	220	72.8	82	27.2		389	30.6	287	73.8	102	26.2	
	Only resistance exercise	150	11.7	104	69.3	46	30.7		49	3.8	34	69.4	15	30.6	
	Both	162	12.6	102	63.0	60	37.0		54	4.2	36	66.7	18	33.3	
	No	672	52.3	484	72.0	188	28.0		781	61.4	555	71.1	226	28.9	
Age								<0.001							0.125
	19~39	24	1.9	17	70.8	7	29.2		22	1.7	13	59.1	9	40.9	
	40~49	114	8.9	94	82.5	20	17.5		72	5.7	58	80.6	14	19.4	
	50~59	258	20.1	200	77.5	58	22.5		207	16.3	154	74.4	53	25.6	
	over 60	890	69.2	599	67.3	291	32.7		972	76.4	687	70.7	285	29.3	
Marital status								0.330							0.546
	Married or Cohabiting	1095	85.1	781	71.3	314	28.7		731	57.4	529	72.4	202	27.6	
	Else	191	14.9	129	67.5	62	32.5		524	41.2	383	73.1	159	30.3	
Educational level								0.220							0.506
	High	284	22.1	207	72.9	77	27.1		89	7.0	61	68.5	28	31.5	
	Middle	421	32.7	306	72.7	115	27.3		249	19.6	185	74.3	64	25.7	
	Low	581	45.2	397	68.3	184	31.7		935	73.4	666	71.2	269	28.8	
Income								0.003							0.405
	High	277	21.5	216	78.0	61	22.0		163	12.8	123	75.5	40	24.5	
	Middle	621	48.3	439	70.7	182	29.3		571	44.9	411	72.0	160	28.0	
	Low	388	30.2	255	65.7	133	34.3		539	42.3	378	70.1	161	29.9	
Occupation								0.025							0.114
	White-collar	208	16.2	155	74.5	53	25.5		60	4.7	45	75.0	15	25.0	
	Pink-collar	89	6.9	68	76.4	21	23.6		144	11.3	110	76.4	34	23.6	
	Blue-collar	472	36.7	345	73.1	127	26.9		228	17.9	173	75.9	55	24.1	
	Housewife or Inoccupation	517	40.2	342	66.2	175	33.8		841	66.1	584	69.4	257	30.6	
Region								0.120							0.911
	Metropolitan city	639	49.7	439	68.7	200	31.3		615	48.3	442	71.9	173	28.1	
	Rural	647	50.3	471	72.8	176	27.2		658	51.7	470	71.4	188	28.6	
BMI								0.258							0.351
	Obesity	597	46.4	426	71.4	171	28.6		629	49.4	462	73.4	167	26.6	
	Normal	672	52.3	475	70.7	197	29.3		626	49.2	438	70.0	188	30.0	
	Low weight	17	1.3	9	52.9	8	47.1		18	1.4	12	66.7	6	33.3	
Drinking								0.412							0.451
	Frequently	726	56.5	503	69.3	223	30.7		184	14.5	129	70.1	55	29.9	
	Occasionally	214	16.6	155	72.4	59	27.6		338	26.6	251	74.3	87	25.7	
	None	346	26.9	252	72.8	94	27.2		751	59.0	532	70.8	219	29.2	
Smoking								0.053							0.317
	Current smoker	374	29.1	282	75.4	92	24.6		52	4.1	36	69.2	16	30.8	
	Ex-smoker	695	54.0	475	68.3	220	31.7		56	4.4	45	80.4	11	19.6	
	None	217	16.9	153	70.5	64	29.5		1165	91.5	831	71.3	334	28.7	
Management of Blood Glucose								<0.001							<.0001
	Non-medication	133	10.3	95	71.4	38	28.6		128	10.1	99	77.3	29	22.7	
	Only medication	987	76.7	695	70.4	292	29.6		985	77.4	702	71.3	283	28.7	
	Only insulin	11	0.9	9	81.8	2	18.2		16	1.3	15	93.8	1	6.3	
	Both medication and insulin	63	4.9	57	90.5	6	9.5		73	5.7	67	91.8	6	8.2	
	None	92	7.2	54	58.7	38	41.3		71	5.6	29	40.8	42	59.2	
Chronic Disease								0.131							0.048
	Only hypertension	404	31.4	269	66.6	135	33.4		324	25.5	218	67.3	106	32.7	
	Only hyperlipidemia	155	12.1	114	73.5	41	26.5		228	17.9	162	71.1	66	28.9	
	Both	390	30.3	278	71.3	112	28.7		524	41.2	377	71.9	147	28.1	
	None	337	26.2	249	73.9	88	26.1		197	15.5	155	78.7	42	21.3	
Year								0.015							0.596
	2015	214	16.6	66	30.8	148	69.2		197	15.5	51	25.9	146	74.1	
	2016	263	20.5	85	32.3	178	67.7		285	22.4	89	31.2	196	68.8	
	2017	261	20.3	87	33.3	174	66.7		261	20.5	77	29.5	184	70.5	
	2018	267	20.8	56	21.0	211	79.0		260	20.4	67	25.8	193	74.2	
	2019	281	21.9	82	29.2	199	70.8		270	21.2	77	28.5	193	71.5	

BMI, body mass index; HbA1c, glycated hemoglobin.

**Table 2 ijerph-19-03280-t002:** Results of multiple regression analysis to investigate the association between physical exercise and HbA1c levels.

Variables		Male				Female			
		HbA1c Level < 6.5%				HbA1c Level < 6.5%			
		OR	95% CI			OR	95% CI		
Physical exercise									
	only walking exercise	0.87	(0.60–1.26)			0.91	(0.65–1.26)		
	only resistance exercise	1.20	(0.74–1.95)			1.01	(0.47–2.15)		
	Both	1.85	(1.17–2.92)			1.08	(0.54–2.17)		
	None	1.00				1.00			
Age									
	19~39	1.00				1.00			
	40~49	0.35	(0.11–1.17)			0.27	(0.09–0.82)		
	50~59	0.66	(0.22–2.04)			0.70	(0.25–1.95)		
	over 60	0.81	(0.27–2.44)			0.60	(0.21–1.68)		
Marital status									
	Married or Cohabiting	1.12	(0.74–1.71)			1.13	(0.82–1.54)		
	Else	1.00				1.00			
Educational level									
	High	1.03	(0.64–1.67)			1.58	(0.83–2.99)		
	Middle	0.84	(0.59–1.20)			0.89	(0.56–1.42)		
	Low	1.00				1.00			
Income									
	High	0.44	(0.27–0.72)			0.68	(0.40–1.16)		
	Middle	0.65	(0.45–0.94)			0.81	(0.57–1.14)		
	Low	1.00				1.00			
Occupation									
	White-collar	1.00				1.00			
	Pink-collar	0.82	(0.41–1.65)			1.11	(0.45–2.74)		
	Blue-collar	0.73	(0.43–1.24)			0.96	(0.40–2.26)		
	Housewife or Inoccupation	0.77	(0.45–1.34)			1.45	(0.66-3.19)		
Region									
	Metropolitan city	1.36	(1.00–1.84)			0.94	(0.70–1.27)		
	Rural	1.00				1.00			
BMI									
	Obesity	1.00				1.00			
	Normal	0.76	(0.56–1.04)			1.20	(0.89–1.62)		
	Low weight	1.93	(0.56–6.63)			1.53	(0.53–4.45)		
Drinking									
	Frequently	1.00				1.00			
	Occasionally	0.59	(0.39–0.89)			0.86	(0.53–1.39)		
	None	0.74	(0.51–1.08)			0.97	(0.61–1.53)		
Smoking									
	Current smoker	1.00				1.00			
	Ex-smoker	1.41	(0.99–2.03)			0.54	(0.19–1.56)		
	None	1.10	(0.69–1.76)			0.71	(0.33–1.56)		
Management of Blood Glucose									
	Non-medication	4.65	(1.37–15.81)			3.47	(1.17–10.27)		
	Only medication	5.17	(1.63–16.38)			4.73	(1.78–12.59)		
	Only insulin	7.51	(1.17–48.23)			2.37	(0.25–22.98)		
	Both medication and insulin	1.00				1.00			
	None	8.94	(2.48–32.18)			18.97	(6.11–58.88)		
Chronic Disease									
	Only hypertension	1.07	(0.73–1.55)			1.09	(0.75–1.57)		
	Only hyperlipidemia	0.98	(0.58–1.66)			1.11	(0.75–1.66)		
	Both	1.00				1.00			
	None	0.83	(0.55–1.25)			0.52	(0.32–0.85)		

BMI, body mass index; HbA1c, glycated hemoglobin.

**Table 3 ijerph-19-03280-t003:** Results of subgroup analysis stratified by independent variables related to health status and behavioral pattern.

Variables		Male													Female												
		Physical Exercise													Physical Exercise												
		None	Only Walking				Only Resistance				Both				None	Only Walking				Only Resistance				Both			
		OR	OR	95% CI			OR	95% CI			OR	95% CI			OR	OR	95% CI			OR	95% CI			OR	95% CI		
BMI																											
	Obesity	1	0.97	(0.57–1.66)			1.28	(0.66–2.45)			2.12	(1.06–4.25)			1	0.89	(0.53–1.47)			0.68	(0.21–2.23)			2.89	(1.15–7.27)		
	Normal	1	0.9	(0.54–1.50)			1.15	(0.56–2.38)			1.76	(0.96–3.22)			1	0.91	(0.57–1.45)			1.09	(0.40–2.95)			0.44	(0.15–1.27)		
	Low weight	1													1												
Drinking																											
	Frequently	1	0.84	(0.52–1.35)			1.49	(0.77–2.88)			2.33	(1.29–4.19)			1	0.5	(0.18–1.34)			0.26	(0.05–1.41)			0.22	(0.05–0.98)		
	Occasionally	1	0.73	(0.24–2.27)			0.76	(0.19–3.03)			1.83	(0.67–4.99)			1	1.73	(0.87–3.47)			5.42	(1.39–21.11)			3.2	(0.97–10.59)		
	None	1	1.09	(0.49–2.40)			0.72	(0.31–1.68)			1.19	(0.39–3.69)			1	0.84	(0.54–1.30)			0.35	(0.12–0.96)			1.03	(0.43–2.44)		
Smoking																											
	Current smoker	1	0.73	(0.36–1.45)			1.36	(0.51–3.64)			1.73	(0.60–5.04)			1												
	Ex–smoker	1	0.98	(0.59–1.64)			1.26	(0.68–2.35)			1.73	(0.97–3.09)			1												
	None	1	0.77	(0.32–1.86)			0.7	(0.22–2.18)			2.04	(0.70–5.92)			1	0.91	(0.64–1.28)			1.04	(0.46–2.32)			1.02	(0.46–2.27)		
Management of Blood Glucose																											
	Non–medication or etc	1	0.71	(0.19–2.58)			0.96	(0.16–5.57)			3.22	(0.84–12.30)			1	0.19	(0.04–0.93)			0.04	(0.00–1.86)			1.05	(0.15–7.64)		
	Only medication	1	0.82	(0.54–1.24)			1.24	(0.70–2.19)			1.6	(0.94–2.73)			1	0.95	(0.66–1.38)			1.06	(0.46–2.41)			1.29	(0.59–2.83)		
	Only insulin	1	1	(0.06–16.21)			1				1				1	1	(0.02–51.80)							1	(0.02–51.80)		
	Both medication and insulin	1					14.69	(0.55–389.17)							1												
	None	1	1.08	(0.18–6.32)			0.58	(0.09–3.53)			1.66	(0.23–12.03)			1	11.16	(1.01–123.51)			1.35	(0.06–28.38)			18.87	(0.52–681.57)		

OR, odds ratio; CI, confidence interval.

## Data Availability

The dataset analyzed in the present study is publicly accessible. Available online: https://knhanes.kdca.go.kr (accessed on 3 February 2022).

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
