# Peer review of "Association between Physical Exercise and Glycated Hemoglobin Levels in Korean Patients Diagnosed with Diabetes"

_ijerph, 2022, doi:10.3390/ijerph19063280_

Round 1
Reviewer 1 Report
This manuscript reads well. Overall, methods and results are clearly presented. Important points related to interpretation of the results are included in the Discussion though I do have some suggestions for other points to consider. The results of this study have important implications for the treatment of diabetes and development of future studies related to exercise and management of diabetes. I have a few points that need clarification.
Introduction
Since resistance and walking/aerobic exercise were distinguished in this study, a little more background and rationale, particularly for resistance exercise, are needed.
Methods
Please provide more information about the questions used to ascertain exercise. Additionally, were they administered by interview, or by a questionnaire that the participant completed by themselves.
Lines 94-99: Four groups are described here. However, in the tables under “Management of glucose”, there is a category called “None”. Who are these individuals and how do they differ from the ‘Non-medication’ group?
Discussion
Lines 163-168: Do these studies separate resistance exercise from walking/aerobic exercise? It would be helpful to discuss these studies in a bit more detail in relation to the results of the current study.
A little more discussion of the results in relation to other studies is needed. What have other studies observed? Only one study is briefly mentioned in Lines 194-196. How are results of this study similar and/or different?
It seems that diet should be addressed somehow in the Discussion. Diet is such a critical part of management of blood glucose that it seems odd to not mention it, perhaps as a limitation?
Reviewer 2 Report
The paper by Yun et al. analyzed the Association Between Physical Exercise and Glycated Hemoglobin Levels in Korean Patients Diagnosed with Diabetes. The authors examine the Korean national health and nutrition examination data from 2015–2019 and used many statistical analyses to determine the relationship between this association. The title and abstract are appropriate for the content of the text.
The introduction is too wordy and doesn’t introduce the topic presented in this paper. For example, the introduction should describe the glycated hemoglobin functions and physical exercise, Previous research in the field of diabetes, the benefit of exercise in the control of glycemia, glycosylated hemoglobin A1c values have been considered as important for what reason, how this exercise is important in this control.
Results
Table 1 indicated the general characteristics of the population, but the methods don’t indicate how this data was obtained. The methods indicated that was an analysis of previous o current data obtained from 2015 to 2019. It is confusing this information.
The following analysis was made by analyzing a full 4-year trial or just one week?
How was determined bias in the analysis? For example, self-reported data.
How was measured HbA1c levels? During exercise, after, or it is a 4-year data.
Round 2
Reviewer 1 Report
Thank you for your attention to comments and suggestions.
Reviewer 2 Report
The authors clarified the questions raised during my previous review. I think this paper is an excellent addition to the research on the association between physical exercise and glycated hemoglobin levels that shows that conclude resistance exercise may contribute to the management of HbA1c levels in patients with diabetes.